# Cord-Blood-Stem-Cell-Derived Conventional Dendritic Cells Specifically Originate from CD115-Expressing Precursors

**DOI:** 10.3390/cancers11020181

**Published:** 2019-02-05

**Authors:** Maud Plantinga, Colin G. de Haar, Ester Dünnebach, Denise A. M. H. van den Beemt, Kitty W. M. Bloemenkamp, Michal Mokry, Jaap Jan Boelens, Stefan Nierkens

**Affiliations:** 1Laboratory of Translational Immunology, University Medical Center Utrecht, 3584 XC Utrecht, The Netherlands; m.c.plantinga-2@umcutrecht.nl (M.P.); E.dunnebach-2@umcutrecht.nl (E.D.); d.a.m.vandenbeemt@umcutrecht.nl (D.A.M.H.v.d.B.); boelensj@mskcc.org (J.J.B.); 2Cell Therapy Facility, Pharmacy Department, University Medical Center Utrecht, 3584 XC Utrecht, The Netherlands; c.g.dehaar@umcutrecht.nl; 3Division Woman and Baby, Department of Obstetrics, University Medical Center Utrecht, Birth Center Wilhelmina’s Children Hospital, 3584 EA Utrecht, The Netherlands; K.W.M.Bloemenkamp@umcutrecht.nl; 4Division of Pediatrics, University Medical Center Utrecht, 3584 EA Utrecht, The Netherlands; M.Mokry@umcutrecht.nl; 5Blood and Marrow Transplantation Program, Princess Máxima Center for Pediatric Oncology, 3584 CS Utrecht, The Netherlands; 6Stem Cell Transplant and Cellular Therapies, Department of Pediatrics, Memorial Sloan Kettering Cancer Center, 1275 York Avenue, New York, NY 10065, USA

**Keywords:** dendritic cell, vaccine, cord blood, cancer, DC subsets, DC precursor

## Abstract

Dendritic cells (DCs) are professional antigen-presenting cells which instruct both the innate and adaptive immune systems. Once mature, they have the capacity to activate and prime naïve T cells for recognition and eradication of pathogens and tumor cells. These characteristics make them excellent candidates for vaccination strategies. Most DC vaccines have been generated from ex vivo culture of monocytes (mo). The use of mo-DCs as vaccines to induce adaptive immunity against cancer has resulted in clinical responses but, overall, treatment success is limited. The application of primary DCs or DCs generated from CD34^+^ stem cells have been suggested to improve clinical efficacy. Cord blood (CB) is a particularly rich source of CD34^+^ stem cells for the generation of DCs, but the dynamics and plasticity of the specific DC lineage development are poorly understood. Using flow sorting of DC progenitors from CB cultures and subsequent RNA sequencing, we found that CB-derived DCs (CB-DCs) exclusively originate from CD115^+^-expressing progenitors. Gene set enrichment analysis displayed an enriched conventional DC profile within the CD115-derived DCs compared with CB mo-DCs. Functional assays demonstrated that these DCs matured and migrated upon good manufacturing practice (GMP)-grade stimulation and possessed a high capacity to activate tumor-antigen-specific T cells. In this study, we developed a culture protocol to generate conventional DCs from CB-derived stem cells in sufficient numbers for vaccination strategies. The discovery of a committed DC precursor in CB-derived stem cell cultures further enables utilization of conventional DC-based vaccines to provide powerful antitumor activity and long-term memory immunity.

## 1. Introduction

Dendritic cells (DCs) are able to elicit strong T-cell responses, which designates them as excellent candidates for vaccine strategies. The commonly used approach to generate DCs for immunotherapy is in vitro differentiation of peripheral-blood-derived monocytes (mo) to mo-DCs in the presence of granulocyte–macrophage colony-stimulating factor (GM-CSF) and/or interleukin (IL)-4 [1,2]. Although vaccination with mo-DCs is feasible and safe, clinical efficacy in terms of overall survival is still limited [3,4]. Recent technological advances have enabled the isolation of reasonable numbers of primary DCs through apheresis that can be antigen loaded for use as a vaccine. The first trials using autologous primary DCs not only confirmed feasibility and safety [5,6,7,8,9] but also showed improved progression-free survival and effective antitumor responses [6]. Although no randomized studies with mo-DCs vs. primary DCs have been performed, these data suggest the preference for using primary DCs instead of mo-DCs for vaccination. However, the low number of primary DCs and the demanding procedure to isolate them from peripheral blood remains an important limitation of general application. Another option is to differentiate DCs from CD34^+^ hematopoietic stem cells, which can be isolated from blood, bone marrow, adipose tissue, or umbilical cord blood (CB) [10,11,12]. Recent technological advances have led to significant insights into DC ontogeny [13,14,15,16] and the unique characteristics of various DC subsets [17,18,19]. Dendritic cells derive from monocyte and DC progenitors (MDPs), which give rise to either common monocyte precursors (cMoPs) or common DC precursors (CDPs). cMoPs develop into monocytes and subsequently into mo-DCs under inflammatory conditions, while CDPs progress to plasmacytoid DCs (pDCs) and conventional DCs (cDCs). The latter differentiate into cDC1 and cDC2, expressing Clec9A and CD1c, respectively. Adding to the complexity of understanding DC ontogeny, tracing studies using barcoding and adoptive transfer studies in humanized mice revealed that next to CDPs, multilymphoid progenitors (MLPs) substantially contribute to cDC development [13,20]. The best DC candidate to induce antitumor immunity has not been elucidated.

We previously showed the feasibility of generating a good manufacturing practice (GMP)-compliant DC vaccine from a very limited number of CB CD34^+^ cells [21]. By expanding and differentiating CB-derived stem cells, we were able to generate 4–6 × 10^6^/mL cells from 5 × 10^4^/mL stem cells. This protocol allows the application of CB-derived DC (CB-DC) vaccination. CB units can be frozen in two fractions: one part (4/5) used for hematopoietic cell transplantation (HCT), the other (1/5) used to generate DCs. With this, we have the unique opportunity to educate the reconstituting immune system after HCT in vivo with CD34^+^-derived DCs generated ex vivo from the same CB as used for transplantation to prevent relapses in leukemia patients treated with a CB HCT. We hypothesized that the success to prevent relapse most likely depends on the immune stimulatory capacity of the DCs to induce long-lasting immunity. Knowledge of CB-DC development and functional specification is required to be able to manipulate and select for the most potential DCs. In the present study, we performed an in-depth analysis into DC development from CB stem cells and identified the progenitor accountable for the generation of GMP-compliant DCs.

## 2. Results

### 2.1. CB-DCs Arise from CD115^+^ Myeloid Progenitors

CD34^+^ stem cells were isolated from CB and expanded and differentiated according to the previously described GMP protocol, providing CB-derived cells referred to as “bulk” (Figure 1A). The cells were analyzed by flow cytometry after expansion (day 7) and differentiation (day 14). Besides a clear DC population (CD11c^+^HLA-DR^+^), the culture contained a fraction of cells that did not express CD11c and/or HLA-DR after differentiation (Figure 1B). After sorting these non-DCs followed by another differentiation cycle, these cells subsequently acquired a DC phenotype (Appendix A), indicating that these cells represent myeloid progenitors still present after one week in differentiation medium. To study these myeloid progenitors in the CB culture in more detail, we isolated different populations after expansion on day 7 (Figure 1C): CD34^+^ (8%) and CD115/CD123/CD45RA-expressing cells (26%), referred to as CD115^+^ and CD117^hi^ (<3%). The variable percentage of CD34^+^ and CD115^+^ within several donors is shown in Figure 1D. The remaining population expressed little CD135 and CD123, referred to as “negative”. Sorted cells were differentiated and, again, CD34^+^ cells gave rise to 10–20% DCs (CD11c^+^HLA-DR^+^) (i.e., similar to the original CD34^+^ culture). Remarkably, CD115^+^ cells generated almost exclusively DCs (75–95%) (Figure 1E), whereas the negative and CD117^hi^ populations did not generate an adequate amount of DCs (Appendix A). These results show that CB-derived DCs originated specifically from CD115^+^ cells in these cultures.

### 2.2. Uptake and Processing by CD115-DCs

To assess the functional capacities of the CD115-derived DCs, antigen uptake was assessed by adding BSA-FITC to the culture for 30 min in different concentrations. CD115-DCs were compared to bulk DCs to confirm their resemblance. The antigen was taken up dose dependently by CD11c^+^HLA-DR^+^ DCs in both the CD115-derived culture as well as in the bulk culture (Figure 2A). In addition, CD115-derived DCs showed the equivalent processing capacity of BSA-DQ overtime compared with CD11c^+^HLA-DR^+^ DCs in the unsorted cultures (Figure 2B). In summary, DCs generated from CD115 progenitors had the capacity to take up and process protein antigens similar to that of DCs from the bulk culture.

### 2.3. Maturation and Migration of CD115-DCs

Next to the classical DC hallmarks of uptake and processing capacity, maturation is pivotal for a strong T-cell response. Therefore, CD115-derived DCs were matured for 24 h with a classical GMP-compliant cytokine mix containing IL-1β, IL-6, TNFα, and PGE2. Bulk DCs were used as a reference. Although the percentage of DCs expanded in these final 24 h (light grey (immature) compared to darker grey (mature)), only up to 13% (8%–16%) within the bulk culture expressed CD83 (black) after maturation. In the CD115-DCs, the percentage CD83^+^ DCs increased significantly up to 40% (21%–66%) of DCs (Figure 3A). In addition, CD115-DCs (dotted line) showed higher fluorescence intensity of CD40, CD80, HLA-ABC, and PD-L1 gated within the HLA-DR^+^CD11c^+^ (DC) population (Figure 3B). However, CD115-DCs produced similar levels of cytokines such as IL-10, RANTES, and IP-10 (Figure 3C). Subsequently, we analyzed the expression of CCR7, which is an important chemokine necessary for migration towards CCL19 or 21-positive-T-cell-rich lymph nodes. Within the DC population, CCR7 surface levels were higher in the CD115-DCs compared with the bulk DC. Next, we analyzed the migratory capacity of matured DCs using a transwell system in the presence or absence of CCL19 (Figure 3D). Although not significant, more CD115-DCs migrated towards CCL19 in comparison to the bulk CB-DCs (Figure 3E,F). In conclusion, not only did the CD115 culture generate a high percentage of DCs, the CD115-DCs matured upon GMP-compliant cytokine mix stimulation and had a substantial capacity to migrate.

### 2.4. T-Cell Activation by CD115-DCs

To test if these mature DCs had a strong ability to stimulate T cells, we cocultured the CD115-DCs and bulk DCs with T cells in an allogenic mixed leukocyte reaction. CD115-DCs showed a similar degree of allostimulatory capacity compared with bulk DCs for both CD4 as well as CD8 CB T cells (Figure 4A). To test the antigen-presenting capacity, CB-DCs from both cultures were matured and pulsed overnight with Wilm’s tumor 1 (WT1) antigen. After 24 h, the CD83^+^ DCs from both cultures were sorted and subsequently cocultured for 5 h with WT1-specific T cells in the presence of brefeldin A. LAMP-1 expression and IFNγ and TNFα production by T cells were increased when stimulated by WT1-loaded DCs from both cultures (Figure 4B). Altogether, the CD115 culture generated a high proportion of DCs which expressed high levels of costimulatory signals. CD115-DCs were highly migratory and possessed strong T-cell stimulatory potential.

### 2.5. Identification of a Specific Progenitor

Next, we set out to define the type of DCs and performed RNA sequencing using flow cytometry based sorted CD115^+^ precursors or well-described monocytes isolated from CB using CD14^+^ magnetic beads. Principal component analysis (PCA) analysis clearly distinguished CD115^+^ cells from monocytes (Figure 5A). Subsequently, we compared CD115-DCs and mo-DCs on a genetic level using PCA with RNA sequencing data. Mo-DCs were generated from CB to compare both cultured cells in order to reduce the differences created by culture techniques. The genetic makeup clearly separated CD115-DCs from mo-DCs, similar to CD115 precursor separation from monocytes (Figure 5B). Next, myeloid genes based on prior knowledge from previous DC studies were analyzed. In the differentiated DCs, a clear pattern was seen regarding cDC genes (e.g., IRF4, FceR1, and CLEC10A were predominantly expressed by CD115-DCs). However, in the precursors, no clear distinction was observed (Figure 5C). For a more in-depth analysis regarding these differences in CD115-DCs and mo-DCs, a heatmap was generated regarding the significantly different expressed genes between the two populations, which showed that 2103 genes in CD115-DCs and 1899 genes in mo-DCs were upregulated. From these, the top 500 genes were used for gene ontology (GO) analysis to highlight the biological processes upregulated in the two types of DCs. In the CD115-DCs, migratory and immune-response-related genes were detected, while in mo-DCs, cell homeostasis was upregulated (Figure 5D). Overall, we noted that CD115-DCs were quite different compared to mo-DCs and had a migratory and inflammatory profile. Therefore, we next questioned if they resemble cDCs to a higher extent. Recent advances have led to a more precise identification of cDC progenitors in CB, bone marrow, or even peripheral blood. We tested the protein expression of described cDC precursors (Figure 6A). CD115 progenitors expressed CD33, CD123, and CX3CR1 like pre-cDCs, however they lacked CD303.

We determined the expression of typical myeloid markers using RNA sequence data (Figure 6B) and flow cytometry (Figure 6C) on differentiated DCs to further confirm the type of DC. Primary cDC1s have been described to express CD141^hi^, CLEC9A^+^, and XCR1, while cDC2s predominantly express CD1c^+^, CD123 expressed on pDCs (but also on precursors), and CD1a on dermal DCs. CD115-DCs showed FceR1, CLEC10A, CD1c, and CD141 expression but no or very low CD123 and CD1a expression, some XCR1 expression, and intermediate CLEC9A levels (Figure 6C). Additionally, BubbleGUM-based comparison of our data with the gene sets classified by Bakdash et al. [22], Balan et al. [12], and Segura et al. [23] (noted in the Figure 1, Figure 2 and Figure 3 respectively) showed particular overlapping genes of CD115-DCs with cDC2s (BDCA1), while CB-derived mo-DCs were enriched in macrophages (Figure 6D). When we compared our dataset to the gene sets recently generated by Villani et al. [15], the CD115-DCs were again mostly enriched in the cDC2_A gene set. These data strongly suggest that CD115-DCs resemble cDCs developing in these GMP-compliant cultures from CD34-derived cDC precursors and are clearly different from mo-DCs.

## 3. Discussion

DCs are potent candidates for vaccination since they exploit the patient’s own immune system to prevent relapses. CB is a rich source to provide CD34^+^ stem cells for the generation of CB-derived DCs. Previously, we generated DCs from CB stem cells, but the DC lineage and reference to primary DC subsets has not been determined. Understanding the development of CB-DCs is a prerequisite to being able to direct the CB culture towards the most potent DC subset. This study identified that CD115^+^ precursors exclusively differentiate towards DCs within the CB-derived stem cell cultures (CD115-DCs). The CD115-DCs were subjected to functional assays and compared to the bulk CB-DC cultures, which displayed cellular heterogeneity. The CD115-DCs were able to take up and process antigen, migrate to a CCL19 gradient, and induce strong T-cell activation in both an antigen-dependent and -independent (allo) fashion. CD115-DCs did perform as well as the CB-DCs, suggesting no synergy of different cell populations in the CB culture.

CD115 is expressed on both MDPs and monocytes [16] and possibly on CDPs, as described in mice [24,25,26]. Considering the almost pure DC population generated from CD115^+^ cells, we anticipated that CD115^+^ cells are committed progenitors. However, the CD115^+^ cells do not cluster together with monocytes in the PCA plots but exhibit substantial overlap with pre-cDCs, both expressing CD45RA, CD123, CX3CR1, and CD33. This suggests that CD115^+^ precursors predominantly comprise a population of CDPs generating cDCs. No discriminative marker or profile for cDCs has been described in humans despite the urge and extensive analysis [27]. The CD115-DCs express CD1c and partly CD209 (Appendix A). The latter is described specifically for mo-DCs, upregulated in an IL-4-dependent manner [28]. Indeed, we detected loss of CD209 expression in the absence of IL-4 in the CB culture (Appendix A). Of note, only a small part of the CD115^+^ cells express CD14, which could imply that some mo-DCs differentiated from monocytes in these cultures. However, after differentiation and maturation, these cells lose their CD14^+^ expression, while mo-derived cells (inflammatory DCs) are identified by their retained CD14 expression [23]. Although depending on culture methods or environmental cues, CD14 expression can be modulated on the cell surface [29] and is therefore solely not suitable to discriminate mo-DCs from cDC2s.

The development described herein of CB cells is limited to the use of specific cytokines in a two-step protocol in the absence of stromal cells, which produces sufficient numbers of DCs for vaccination but may limit the generation of alternative DC subsets. During expansion, IL-3 and SCF are used to maintain stem cell proliferative capacity [30], which increases cell numbers, but CD34 expression is lost over time, as previously reported [31]. When human serum in these cultures is replaced by fetal calf’s serum (FCS), CLEC9A^+^DCs can be obtained from the CD115^+^ precursors (Appendix A), as previously described by Poulin et al. [19]. This indicates that CD115s are DC progenitors but not yet committed to one specific subset. This could be explained through heterogeneity: with increasing technology to study progenitors at the single-cell level, increased knowledge of DC progenitors and their heterogeneity is revealed. It was found, for instance, that pDCs contained a pre-cDC population expressing CD123 and CD303, which was excluded previously [14]. Furthermore, heterogeneity was seen within the cDC2 population, resulting in the complexity of discriminating cDC2s from mo-DCs with overlapping phenotypes and gene signatures [15,32]. However, RNA levels of *KLF4* are more highly expressed in CD115-DCs compared with mo-DCs. KLF4, previously assigned to be required for monocyte development and macrophage polarization [33], has been more recently reported, at least in mice, as a potential candidate for cDC2 development, leading to Th2 immunity [34]. This suggests that gene expression levels of both cDCs and mo-DCs transcription factors is not per se explained by heterogeneity within a population but rather an unidentified functional specification. The identification of DC subsets is complicated, although the phenotype of DC subsets is largely aligned in tissues and between species [27]. Most studies are performed on primary cells and not correlated to in vitro cultured DCs. GM-CSF has been described to induce CD1 family members [35], complicating the discrimination by flow cytometry of cDC2s (CD1c+) from mo-DCs in culture. However, gene analysis strongly separates CD115-DCs from CB-derived mo-DCs.

CD115-DCs were matured with cytomix, classically used to mature mo-DCs [36]. TLR stimuli might mature DCs differently (Appendix A), and a more pronounced maturation might be expected if TLR agonists are combined [37,38]. CXCL9 has been produced by murine cDC1 in a tumor model [39], recruiting CD8 T cells within the tumor microenvironment. However, we hardly detected CXCL9, perhaps due to the single TLR stimulus. R848, in addition to poly I:C, has been shown to enhance CXCL9 production in cDC1 [40]. However, many TLR stimuli are not yet GMP-grade available and therefore the analyses in this paper focused on cytomix.

We showed that CB-derived stem cells can extensively be expanded and differentiated into DCs which genetically, phenotypically, and functionally resemble cDC2 primary cells. These cells may provide a potent alternative strategy to stimulate antitumor immunity and prevent relapses in HCT patients.

## 4. Materials and Methods

### 4.1. CB Collection and CD34 and CD14 Isolation

Umbilical cord blood was collected after informed consent was obtained according to the Declaration of Helsinki. The ethics committee of the University Medical Center Utrecht approved these collection protocols. The protocol number of the ethical committee within the UMC Utrecht is TC-bio 15-345. CB mononuclear cells were isolated from human umbilical CB by density centrifugation over Ficoll-Paque solution (GE Healthcare Bio-Sciences AB, Chicago, IL, USA). CD34^+^ cells were isolated from fresh CB using magnetic bead separation (Miltenyi Biotec, Bergisch Gladbach, Germany) resulting in an 80–95% pure CD34^+^ population after running two columns, as determined with flow cytometry. CD14^+^ cells were isolated from fresh CB using magnetic bead separation (Miltenyi Biotec) resulting in a 95% pure CD14^+^ population.

### 4.2. CB-DC or Mo-DC Culture

For the CB-DCs, the two-step protocol consisted of expansion and differentiation phases. In the expansion phase, 5 × 10^4^ CD34^+^ cells/mL were cultured in X-VIVO 15 supplemented with Flt3L (50 ng/mL), SCF (50 ng/mL), IL-3 (20 ng/mL), and IL-6 (20 ng/mL) for 7 days. After washing, the cells were differentiated at 2 × 10^5^ cells/mL in differentiation medium and X-VIVO 15 containing 5% human AB serum and supplemented with Flt3L (100 ng/mL), SCF (20 ng/mL), GM-CSF (20 ng/mL), and IL-4 (20 ng/mL) for another 7 days [19]. Recombinant cytokines were all obtained from Miltenyi Biotec. To generate control mo-DCs, CB-derived CD14^+^ cells were cultured for 7 days in X-VIVO 15 containing 5% human AB serum and supplemented with IL-4 (25 ng/mL) and GM-CSF (100 ng/mL). After 3–4 days of culture, fresh cytokines were added to the culture.

To induce maturation, cytomix, a combination of IL-1beta, IL-6, and TNFα (all used at 10 ng/mL) and PGE2 (1 µg/mL) from Pfizer, was added to the DCs for 24 h in combination with the differentiation medium. When indicated, CB-DCs or CD115-DCs were stimulated with either CpG (ODN2216; 2 µM), LPS (100 ng/mL), R848 (3 µg/mL), or Poly I:C (30 µg/mL) for 24 h. When indicated, supernatants were analyzed using multiparameter Luminex (BIO-RAD, Hercules, CA, USA). To obtain CD115^+^ DCs, CD115^+^ cells were sorted after 7 days of expansion of CD34^+^ cells. Cells were counted and stained with a progenitor staining mix. After sorting (>95% purity), cells were differentiated at 2 × 10^5^/mL in differentiation medium as described above.

### 4.3. Flow Cytometry

Anti-CD1c (L161), anti-CD3 (UCHT1), anti-CD45RA (HI100), anti-CD115 (9-4D2-1E4), anti-CD116 (4H1), anti-CD117 (104D2), anti-CD123 (6H6), anti-CLEC9A (8F9), anti-CLEC10A (H037G3), anti-CX3CR1 (2A9-1), anti-HLA-DR (L243), anti-PD-L1 (2A3), anti-TNFα (Mab11), and anti-XCR1 (S15046E) were purchased from Biolegend (San Diego, CA, USA). Anti-BDCA-3/CD141 (AD5-14 H 12) and anti-BDCA2/CD303 (AC144) were obtained from Miltenyi Biotec. Anti-CD1a (HI149), anti-CD8α (RPA-T8), anti-CD11c (B-ly6), anti-CD34 (581), anti-CD40 (5C3), anti-CD80 (L307.4), anti-CD83 (HB15e), anti-CD107a/LAMP-1 (H4A3), anti-CD123 (7G3), anti-CD135 (4G8), anti-CD197/CCR7 (3D12), anti-CD209 (DCN46), anti-HLA-ABC (G46-2.6), and anti-IFNγ (4SB4) were purchased from BD Biosciences (San Jose, California,). Anti-CD33 (WM53) was obtained from Sony Biotechnology (Minato, Tokio, Japan). Anti-CD4 (RPA-T4) and anti-FceR1 (AER-37) were obtained from eBioscience (San Diago, CA, USA). Cells were incubated on 4 °C and stained with appropriate antibody combinations. Multiparameter analysis was performed on a FACS LSR Fortessa (BD Biosciences) flow cytometer. At indicated experiments, cell progenitors were sorted using a FACS ARIA III (BD Biosciences) flow cytometer.

Dead cells were excluded by scatter gating. Analysis was performed using FlowJo software (Tree Star, Inc., Ashland, OR, USA).

### 4.4. Uptake and Processing Assay

For uptake, CB-DC culture was incubated with 0.05 or 0.5 μg/mL BSA-FITC (Molecular Probes) for 30 min at 4 °C to measure nonspecific binding or at 37 °C to measure specific uptake. Cells were then washed extensively with ice-cold PBS, 0.1% FCS, and 0.05% NaN_3_ and labeled at 4 °C with the appropriate antibodies. The actual uptake was determined as the geometric mean of FITC^+^ cells within the DC population incubated at 37 °C minus the percentage of FITC^+^ cells incubated at 4 °C.

For analysis of processing by CB-DC cultures, we used DQ Green BSA, a self-quenched dye conjugate of BSA. CB-DCs were incubated with 0.5 μg/mL DQ Green BSA at 4 or 37 °C for 10 min. After extended washes, CB-DCs were stained with the appropriate antibodies after 2, 6, and 18 h.

### 4.5. Transwell Migration Assay

In vitro migration assays were performed using 24 transwell (3 µm pore size) plates (Greiner, Kremsmünster, Germany). In brief, 400,000 CB-DCs in 200 µL of culture medium (X-VIVO 15 with 5% human AB serum) were plated in the upper compartment. Culture medium, either alone or supplemented with 250 ng/mL of CCL19 (R&D systems, Minneapolis, MN, USA), was added to the lower compartment. After 2 h, cells were collected from the lower compartment and analyzed using flow cytometry.

### 4.6. Mixed Leukocyte Reaction

After isolation of CD34 to generate CB-DCs, the resultant CD34− fraction was enriched for T lymphocytes using anti-CD3 magnetic microbeads (Miltenyi). For mixed leukocyte reaction (MLR), allogeneic lymphocytes (1 × 10^6^/mL) were then labeled with cell trace violet (5 µM; Invitrogen) and cocultured with CB-DCs (2 × 10^5^/mL) in a 96-well round-bottom plate (Corning) at a stimulator:responder ratio of 1:5. Unstimulated cell-trace-violet-labeled cells served as negative control. After 4 days, cells were stained with CD3, CD4, and CD8 and analyzed using a FACScanto (BD). T-cell proliferation analysis was performed using the proliferation tool in flowjo (Tree Star, Inc.), providing the division index.

### 4.7. WT1 Antigen Presentation

CB-DCs or CD115-DCs (50,000) were loaded overnight with *WT1 peptivator* (Miltenyi). The following day, CD83+ DCs were sorted from both cultures and cocultured with an HLA-A2-restricted WT1-specific T-cell clone recognizing the WT1_37–46_ epitope (50,000 T cells) at a DC-to-T-cell ratio of 1:1 for 5 h in the presence of Golgi-stop (1/1500; BD Biosciences). T cells alone or T2 cells loaded with/without WT1_37–46_ peptide (Think Peptides; Appendix A) were used as controls. The T cells were subsequently stained for surface markers and, after fixation and permeabilization with the BD fix/perm (BD Biosciences), labeled with anti-IFNγ antibodies, followed by flow-cytometry-based analysis.

### 4.8. RNA Sequence

Total RNA was extracted using RNeasy Mini Kit (Qiagen) according to the manufacturer’s instructions. The concentration of RNA was quantified using a Qubit RNA HS assay and Qubit fluorometer (Thermo Fisher). Polyadenylated messenger RNA was isolated using Poly(A) beads (NEXTflex, San Jose, CA, USA), and sequencing libraries were made using the Rapid Directional RNA-seq kit (NEXTflex). Libraries were sequenced at the Utrecht Sequencing Facility (USEQ) using the Nextseq500 platform (Illumina), which produced single end reads of 75 bp. Reads were aligned to the human reference genome GRCh37 using STAR version 2.4.2a. Picard’s AddOrReplaceReadGroups (version 1.98) was used to add read groups to the binary sequence alignment files, which were sorted using Sambamba (version 0.4.5), and transcript abundances were quantified with HTSeq-count (version 0.6.1p1) using the union mode. Subsequently, reads per kilobase per million reads sequenced (RPKMs) were calculated with the edgeR RPKM function (Appendix A).

Differentially expressed genes were identified using the DESeq2 package with standard settings. Genes with an absolute log_2_ fold change larger than 0.6 and adjusted *p* values of less than 0.1 were considered to be differentially expressed genes.

### 4.9. RNA Sequence Analysis

Transcription factor expression was done on Z-Normalized RPKM values generated from the read counts. Samples were divided over two groups according to sequencing run and Z normalized within their group. A PCA plot was generated using the default PCA function in R. ToppGene was used to analyze Gene set enrichment analysis (GSEA). The BubbleGUM software [41] was used to assess CD115-DC or mo-DC enrichment across myeloid cell types from other datasets. We used the inflammatory DC (infDC) and CD14+ monocyte (MONO) gene set from Bakdash et al. [22] (noted in Figure 6) In addition, we used the MoDC signature, BDCA1 DC signature, macrophages, and CD16+ monocytes gene sets previously generated by Segura et al. [23] (noted in Figure 6) The cDC1 gene set was obtained from Balan et al. [12] (noted in Figure 6) A second BubbleGUM analysis was performed using the gene sets generated by Villani et al. [12,15,22,23]. The detailed composition of all these gene sets is provided in Appendix A.

## 5. Conclusions

DCs excel in elucidating strong T-cell responses and are therefore good candidates for cancer vaccines. DC vaccination has been demonstrated to be safe and has shown immunological responses in a minority of patients treated for a variety of cancers. For improvement in clinical efficacy, there is a strong medical need to improve DC vaccines.

Here, we presented a culture protocol to generate large numbers of potent DCs from CB-derived stem cells, which resemble the blood cDC counterparts. We created a tool not only to study conventional DC biology in more detail but, importantly, to generate sufficient conventional-like DCs for vaccine strategies.

## Figures and Tables

**Figure 1 cancers-11-00181-f001:**
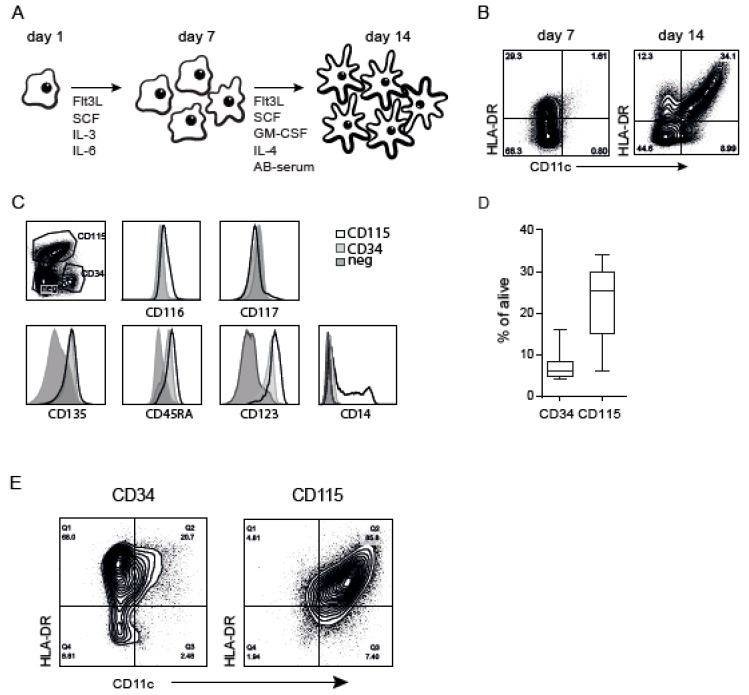
Cord blood (CB)-derived dendritic cell (DC) (CB-DC) culture containing different myeloid progenitors. (**A**) Schematic overview of the culture protocol for the generation of the backbone CB-DC. (**B**) CD11c and HLA-DR expression using flow cytometry after expansion (day 7) and after differentiation (day 14) within the CB culture. (**C**) Identification of DC progenitors after the expansion phase based on CD34, CD115. CD116, CD117, CD135, CD45RA, CD123, and CD14 expression compared between CD115^+^, CD34^+^, and the remaining population (negative). (**D**) Percentage of the presence of the progenitors in the CB culture. (**E**) The expression of CD11c and HLA-DR after differentiating the sorted progenitors in differentiation medium. Data represent at least five independent experiments.

**Figure 2 cancers-11-00181-f002:**
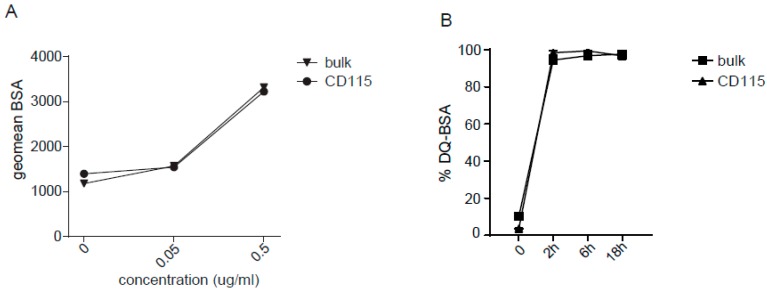
CD115-DCs can take up and process antigen. (**A**) Uptake of increasing doses of BSA-FITC by DCs from the standard or CD115 culture. The geometric mean of FITC^+^ signal taken up at 37 °C is shown minus the FITC^+^ signal detected at 4 °C. (**B**) The processing of BSA-DQ is shown as a percentage of DQ-BSA gated within the DCs from the standard or CD115 culture at indicated time points. The DQ signal seen at 4 °C is subtracted from the DQ^+^ cells at 37 °C. Data represent two independent experiments. The experiments were performed at least three times.

**Figure 3 cancers-11-00181-f003:**
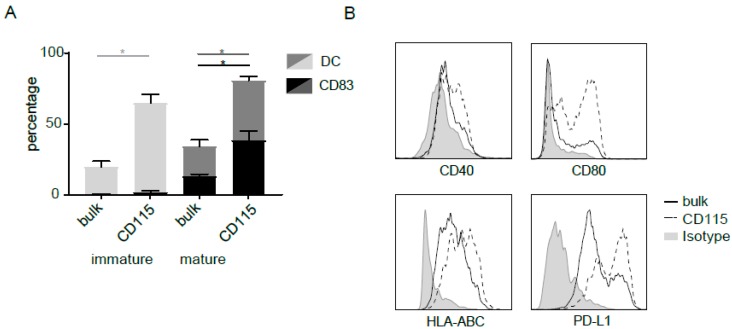
Maturation and migration by CD115-DCs. For flow cytometry analysis, live cells were gated with SSC/FSC, followed by doublet exclusion based on FSC-W/FSC-A. (**A**) Percentages of DCs (HLA-DR+CD11c+) and CD83 at the end of differentiation (day 14) and after an additional 24 h of maturation with cytomix. (**B**) Expression of CD40, CD80, HLA-ABC, and PD-L1 on DCs (live/single/CD11c+HLA-DR+) from the standard (black line) compared to the CD115 culture (dotted line), with an isotype as control (grey). (**C**) Cytokines (IL-10, RANTES, and IP-10) produced by standard or CD115 culture after 24 h of maturation. (**D**) Expression of CCR7 on DCs (live/single/CD11c+HLA-DR+) from the standard (black line) compared to the CD115 culture (dotted line), with an isotype as control (grey). (**E**) Representative migration assay towards CCL19 or medium as a control with DCs from either the bulk or the CD115 culture. (**F**) Quantification of migration with four different donors. Count of migrated mature DCs is shown. When indicated, significance was assessed using two-way ANOVA followed by Kruskal–Wallis multiple comparison (* *p* < 0.05).

**Figure 4 cancers-11-00181-f004:**
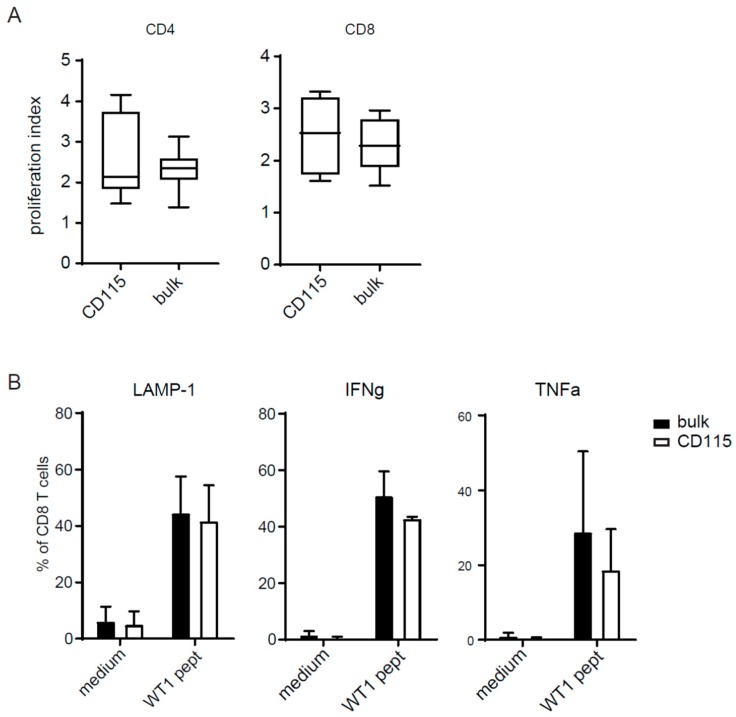
(**A**) T-cell activation was measured in a mixed leukocyte reaction (MLR). Previously isolated CD3 T cells from a different CB donor were thawed and labeled with a cell tracer violet dye. Cells were seeded at 1 × 10^5^ cell/well and stimulated with 2 × 10^4^ cells/well bulk DCs or CD115-DCs for 5 days. Proliferation was measured by FACS and the proliferation index (PI) was calculated using Flowjo. PI is the total number of divisions divided by the number of cells that went into division gated within the CD4 (left) or CD8 (right population). (**B**) Antigen-specific T-cell activation by sorted CD83^+^ DCs pulsed o/n with 6 nmol Wilm’s tumor (WT1) peptivator (Miltenyi Biotec, Bergisch Gladbach, Germany) from the CD115 culture compared to the bulk culture. T-cell activation was measured by their intracellular IFNγ and TNFα and extracellular LAMP-1 expression. A represents four different donors and B from two independent experiments.

**Figure 5 cancers-11-00181-f005:**
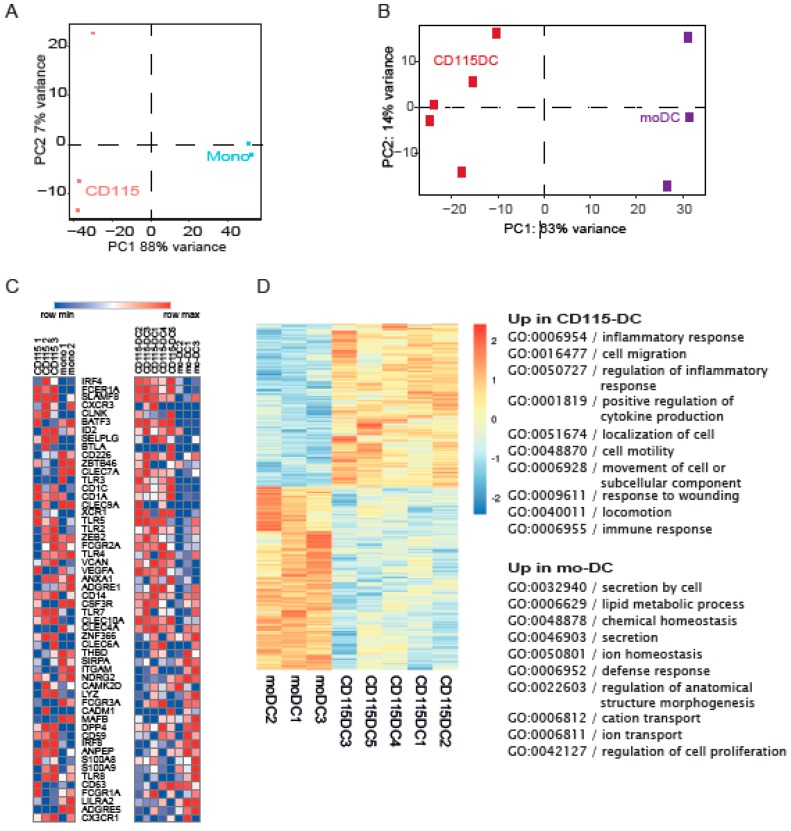
RNA sequence reveals different genetic profiles within the progenitors and DCs. (**A**–**D**) RNA-seq data from monocytes, CD115 progenitors, mo-DCs, and CD115-DCs. (**A**) PCA plot of the CD115 progenitors sorted after 7 days of expansion compared to CD14-isolated monocytes. (**B**) PCA of CD115-DCs compared to IL-4/GM-CSF-cultured CB-derived mo-DCs. (**C**) Heatmap generated comparing genes based on prior knowledge on myeloid development and DC subsets using normalized RPMK from RNA-seq data. (**D**) Heatmap generated from the significantly different genes between mo-DCs and CD115-DCs. Further analysis of the top 500 differentially expressed genes with gene ontology (GO) term analysis using the ToppGene software. The top 10 GO terms of the biological process were upregulated in mo-DCs and CD115-DCs. RNA was isolated from three to five different CB donors. Monocytes and mo-DCs were isolated from two to three CB donors.

**Figure 6 cancers-11-00181-f006:**
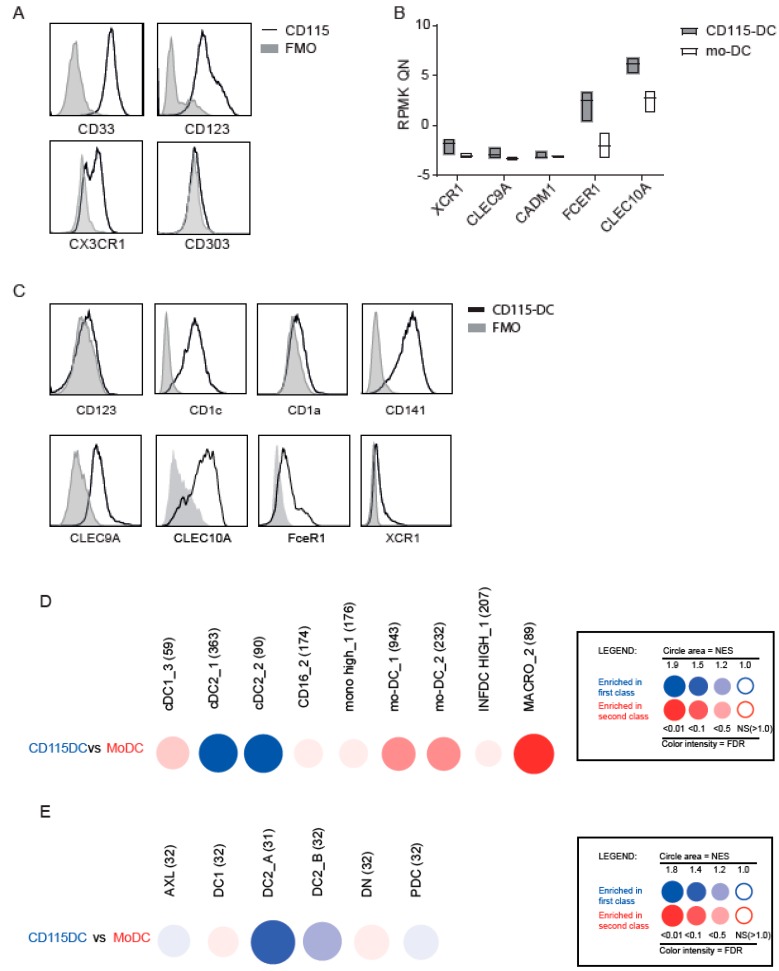
Identification of specific CD115 profile. Transcriptome analysis of CD115(-DC) and mo(-DC) was performed by RNA sequencing of sorted precursors or differentiated DCs generated from CB-derived CD34^+^ stem cells from three to five donors. For FACS analysis, alive cells were gated with SSC/FSC, followed by doublet exclusion based on FSC-W/FSC-A. (**A**) Protein levels of markers expressed on myeloid precursors studied on CD115 progenitors with a fluorescence minus one (FMO) as negative staining control. (**B**) Normalized RNA levels of genes expressed by different DC subsets. (**C**) Protein expression of CD123, CD1a, CD1c, CD141, CLEC9A, CLEC10A, FceR1, and XCR1 within CD115-DCs with FMO control. (**D**,**E**) CD115-DCs and mo-DCs were compared for their relative enrichment in publicly available gene sets using Gene set enrichment analysis (GSEA) through the BubbleGUM software. Results are represented as bubbles in a color matching that of the cell subset in which the gene set was enriched. Stronger and more significant enrichments are represented by bigger and darker bubbles, as illustrated in the legend box of the figure. Specifically, the surface area of the bubbles is proportional to the absolute value of the normalized enrichment score (NES). The color intensity of the dots is indicative of the false discovery rate (FDR) statistical value. The gene sets were defined based on those used by Bakdash et al. [22], Balan et al. [12], Segura et al. [23] (**D**), and Villani et al. [15] (**E**).

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
