# Peer review of "Cord-Blood-Stem-Cell-Derived Conventional Dendritic Cells Specifically Originate from CD115-Expressing Precursors"

_cancers, 2019, doi:10.3390/cancers11020181_

Reviewer 1 Report

The authors try to identify a better source for DC vaccines using in cancer therapy. They find that Cord blood (CB) is a particular rich source of CD34+ stem cells for the generation of DCs and CB-DC exclusively originate from CD115+ progenitors. They also show that these DCs mature and migrate upon GMP-grade stimulation and possess a high capacity to activate tumor-antigen-specific T cells, and then develop a culture protocol to generate conventional DCs from CB-derived stem cells. They suggest the further utilization of CB-DC-based vaccines may provide powerful anti-tumor activity and long-term memory-immunity.

This study is good in DC biology; however, I am not sure whether it is good in cancer research. They did not have a cancer model for treating their DC vaccine and compare it to mo-DC. Thus, I suggest that this manuscript may not be suitable for Cancers and it could be submitted to immunology-related journals.

A minor point is that the conclusion should be re-written. Rationale is not necessary.

Author Response

Reviewer 1:
The authors try to identify a better source for DC vaccines using in cancer therapy. They find that Cord blood (CB) is a particular rich source of CD34+ stem cells for the generation of DCs and CB-DC exclusively originate from CD115+ progenitors. They also show that these DCs mature and migrate upon GMP-grade stimulation and possess a high capacity to activate tumor-antigen-specific T cells, and then develop a culture protocol to generate conventional DCs from CB-derived stem cells. They suggest the further utilization of CB-DC-based vaccines may provide powerful anti-tumor activity and long-term memory-immunity.

This study is good in DC biology; however, I am not sure whether it is good in cancer research. They did not have a cancer model for treating their DC vaccine and compare it to mo-DC. Thus, I suggest that this manuscript may not be suitable for Cancers and it could be submitted to immunology-related journals.

A minor point is that the conclusion should be re-written. Rationale is not necessary.

Answer:

Indeed there is no cancer model tested at this stage, since for the special edition of Cancers; Tumor associated Dendritic cells, we decided to focus on the identification of the source of the DC within the CB-culture to enable optimization for DC vaccination to prevent relapse in refractory patients. Here, we provide a tool to generate GMP-compliant cDC, which can be implemented for cancer treatments. As follow up of this pre-clinical paper on DC presented here, we will focus on functional advances of our GMP-compliant CB-derived DCs in vitro, and additionally in cancer models. We acquired already preliminary data to activate TAA-specific T cells, which subsequently are able to lyse AML lines, but this is beyond the scope of the special edition about Dendritic cells.

Reviewer 2 Report

This manuscript characterized the CD115+ cord blood stem cell derived DC. It was well written and documented. The major concern is that CD11c+DR+ population could include CD14+ monocytes even it was mentioned in the discussion Page 244. It would be very helpful to add the CD14 staining data in Fig. 1C. I will be very curious to see any functional difference comparing Mon-DC with CD115+DC. Lastly the author showed that there were difference between the bulk DC with purified CD115+DC, e.g. migration and maturation, however no MLR difference observed based on proliferation of CD3+ T cells (Fig. 4A), how to explain it? How about using naïve CD4+T cells as responding cells.

Minor comments:

1.       Add more refs. 

In the introduction part, add Shinde P et al  Scientific Report 8: 2018 after (3);

Add Hsu J et al Oncoimmunology 2018; Fromm P et al  Oncoimmunology, 2016 After (4-6);

 add Balan S et al  J Immunol 2014 after (7-8)

2.       Fig 4 legend and introduction line 65/66, format the x10^5 or x10e4 or 10e6 with superscript.

3.       Statistic analysis Fig. 4A.

Author Response

Reviewer 2:
This manuscript characterized the CD115+ cord blood stem cell derived DC. It was well written and documented. The major concern is that CD11c+DR+ population could include CD14+ monocytes even it was mentioned in the discussion Page 244.

Point 1: It would be very helpful to add the CD14 staining data in Fig. 1C.

Answer point 1: We added CD14 staining in figure 1c. A small part within the CD115 population expresses CD14, so we cannot fully exclude monocyte differentiation. However, when we perform transcriptome analysis neither the CD115 resemble monocytes or the CD115-DC the mo-DC.

Point 2: I will be very curious to see any functional difference comparing Mon-DC with CD115+DC.

Answer point 2: We agree with the reviewer that it is interesting to see functional difference comparing mo-DC with CD115-DC, which is currently ongoing. Since this could potentially have a great impact for future DC vaccination strategies we decided here to focus on the type of DC subset we culture in this GMP-compliant protocol to submit to this special edition of Cancers. Now we have established that they are cDC, functional comparison would be the next step.

Point 3: Lastly the author showed that there were difference between the bulk DC with purified CD115+DC, e.g. migration and maturation, however no MLR difference observed based on proliferation of CD3+ T cells (Fig. 4A), how to explain it? How about using naïve CD4+T cells as responding cells.

Answer point 3: In a mixed leukocyte reaction the responder T cells mainly respond to the major histocompatibility antigen (MHC Class I and II) differences between T cells and the stimulatory DC and to a lesser extent co-stimulation. For all the T cell assays we used CB T cells similar to the source used for the DC generation. Cord blood T cells are quite naïve at start of the assay. I included CD4 and CD8 staining and could therefor distinguish between CD4 and CD8 T cells stimulation by the different DC. I changed figure 4A, showing both CD4 and CD8 proliferation, which proliferated in a similar fashion comparing bulk DC to CD115-DC.

Minor comments:

1.       Add more refs. In the introduction part, add Shinde P et al  Scientific Report 8: 2018 after (3); Add Hsu J et al Oncoimmunology 2018; Fromm P et al  Oncoimmunology, 2016 After (4-6);  add Balan S et al  J Immunol 2014 after (7-8)

We thank the reviewer for pointing out these relevant references, which were added to the original paper.

2.       Fig 4 legend and introduction line 65/66, format the x10^5 or x10e4 or 10e6 with superscript.

x10^5 or x10e4 or 10e6 were formatted in the figure legend and introduction.

3.       Statistical analysis Fig. 4A.

Changed the data by separating CD4 and CD8 where the differences are not significant.

Reviewer 3 Report

General comments:

Plantinga et al., demonstrate that the conventional DCs are originated from a CD115+ precursor population and it is a novel information and they are able to demonstrate that the origin of cDCs from CD115+ cells and their similarity with pre-cDC.  Considering the novelty the article should be considered for publication but it requires a major revision with reanalyzing the data as well as performing few additional experiments to unambiguously confirm the identity of DC subset in generated from the CD115+ cells. I think it is important to demonstrate the true identity of these cells subsets to extrapolate their application for cancer immunotherapy.

Major comments

Line 119: Migration assay: The authors trying to compare two different systems with a different base lines.  DCs generated from bulk culture have few CD11c+HLADR+ DCs or MoDCs (10-20%), whereas the CD115 derived cultures are almost 70-90 % of DCs after differentiation. So there is more mature DCs to start with CD115 cultures and authors are comparing that with mixed population of undifferentiated, differentiated and potential precursors of 115 DCs. Hence the authors should compare the migration of Sorted MoDCs from Bulk vs the CD115 DCs to make any solid conclusion. Eventhough the differences are not statistically significant with the current results, the MoDCs may have a better migration if you start with the purified fractions.

Line 123: T cell activation: again authors are doing the same comparison with a bulk population to more homogenous DC subsets.  The current set may not provide an equal probability to interact with the same number of DCs and T cells in two experimental conditions.  So they should start with more homogenous or FACS sorted populations.

Authors FACS staining clearly shows that the CD115+ cells are clec9a+ and CD141+ (Fig 6A  ). It is a clear and specific signature of cDC1 phenotype, even though the author claims the expression of Clec9a is intermediate that is quiet normal with an in vitro generated cDC1. Authors should additional FACS staining on markers including CADM1 (clone 3E1 from MBL), XCR1 (Bio legend), FCERIA, Clec10A to confirm the identity of these subsets.  It is critical to confirm the expression of these genes at protein level with FACS and also demonstrate the expression at transcriptome level from the RNA seq data.

If the cultures are a mixture of cDC1 and cDC2 it will be great approach to distinguish them and separate them for validating the nature of in vitro generated DCs.

The data clearly shows the signature of cDC1 and unfortunately the authors are clearly overlooking those signatures in transcriptome analyses and perform a bubblegum analysis with cDC2 and MoDC signatures (Fig 6D and E).  The data should be reanalyzed with cDC1 gene signature enrichment.( please kindly refer some mentioned  papers performed similar  detailed analysis on such in vitro generated subsets and refer the gene sets).   The analysis should be performed in a non-biased way and why the authors specifically avoiding the gene set specific to the cDC1.

I think major revamp is required on reanalyzing the data and preparing new figures by combining figure 5 and 6 to present more precise and detailed analysis. The authors can prepare a PCA or unsupervised hierarchical clustering by pooling the RNA seq data from the pre and post differentiated subsets to show their difference and similarity. The authors should perform a GSEA or bubble gum analysis on more unbiased way and must include the cDC1 signature.  Authors should present the list of highly expressed genes or differentially expressed in each subsets before and after differentiation and they can illustrate that in the form of heat map. Also it will be interesting to see the heatmap with the top hits to describe the first 10-15 genes to explain the identity of the subsets. That can replace the figure 5B, 5C and 6C.

Additional suggestions: It will be great to include a poly I:C  ( Poly I:C LC), R848 and LPS as maturation stimuli ( both are clinically relevant) and check for the cytokine production. That can be another experiment to confirm the true identity of the subsets based on the cytokine profile including type II IFN production, IL-12, IL-10, CCL9 production etc. They have differential response to each TLR ligands and produce different cytokines ( refer the suggest papers)

Minor comments: 85-95: on different subsets:  The CD116 and CD117 populations are not very distinct subsets from the FACS staining as compared to the CD115 vs CD34 or HLADR vs CD11c. It will be great include an FMO controls in FigS2 to make sure that they are a real population. There are few instants the authors mention that the data not shown. I think most of the current journals do not support such claims and it will be better to include all those data as supplementary figures. (line 243, 255 line 348)

Other comments: It will be great to select references where the DC subsets are well characterized with multiple parameters than few ambiguous surface markers as well as redefining the published information based on the recent literatures. (Refernce 8) The study identifies cDC2 as BDCA1+ CD14+ and there are well defined markers to identify the cDC2 including FCERIA, CLec10 A, BTLA and CD32b etc are more specific markers to identify the cDC2References 10 and 11 clearly shows that the CD123+BDCA2+ are a mixed population of cells and they are not pure pDCs. They also demonstrate that pDCs are poor stimulators of T cells.  So we are not sure the pDCs in this paper is true pDCs or mixture of cells and that is the reason they get a response and other paper fails to show pDCs elicit T cell response. The concern is also applicable for this manuscript and the authors claims the population is more cDC2 based on CD1c expression and they have to define them better by additional markers.  There are publications already demonstrate that the in vitro generated cDC1 and even MoDCs can  expresses high level of CD1c (PMID: 25009205). Some of these publications may be useful for better defining the observed DC subsets in the culture (PMID: 25009205, PMID: 30110645, PMID: 29344995, PMID: 29163495)

Author Response

PDF is attached with the answers to the reviewers questions.

Round  2

Reviewer 1 Report

It is OK now.

Author Response

Thank you for your time to review the manuscript.

Reviewer 3 Report

The authors made significant improvements and the only suggestions is to submit the RNA seq data on GEO (https://www.ncbi.nlm.nih.gov/geo/info/seq.html) for the general use and include the accession number to the paper.  

Author Response

Dear reviewer,

thank you for your time and comments to improve our manuscript. Due to privacy legislation of the European Union, we are not allowed to share raw RNAseq data. But what I can and will do is a supplementary table with the counts of all the genes in the different samples.

Kind regards.